# Enantioselective copper-catalyzed azidation/ click cascade reaction for access to chiral 1,2,3-triazoles

Ling-Feng Jiang[1], Shao-Hua Wu[1], Yu-Xuan Jiang[1], Hong-Xiang Ma[1], Jia-Jun He[1], Yang-Bo Bi[1], De-Yi Kong[1], Yi-Fei Cheng[1], Xuan Cheng [1] & Qing-Hai Deng [1] ✉

Chiral 1,2,3-triazoles are highly attractive motifs in various fields. However, achieving catalytic asymmetric click reactions of azides and alkynes for chiral triazole synthesis remains a significant challenge, mainly due to the limited catalytic systems and substrate scope. Herein, we report an enantioselective azidation/click cascade reaction of *N*-propargyl-β-ketoamides with a readily available and potent azido transfer reagent via copper catalysis, which affords a variety of chiral 1,2,3-triazoles with up to 99% yield and 95% ee under mild conditions. Notably, chiral 1,5-disubstituted triazoles that have not been accessed by previous asymmetric click reactions are also prepared with good functional group tolerance.

1,2,3-Triazoles play a crucial role in drug discovery, biorthogonal chemistry, synthetic chemistry, and materials sciences[1–8]. Since being simultaneously reported by the groups of Sharpless[9,10] and Meldal[11], copper-catalyzed azide-alkyne cycloaddition, also known as click reaction[12–19], has emerged as a highly efficient and biocompatible strategy for synthesizing 1,2,3-triazoles. Despite the well-explored racemic version, the synthesis of chiral triazoles remains challenging mainly due to no $sp^3$ stereogenic center in the triazole skeleton and the linear geometry of the azide and alkyne[20,21]. In this context, Zhou[22–26], Fossey[27–29], Topczewski[30,31], and others[32–39] have employed strategies such as (dynamic) kinetic resolution or desymmetrization to achieve a series of enantioselective copper-catalyzed click reaction of azides and terminal alkynes, which generated enantioenriched 1,4-disubstituted triazoles regioselectively (Fig. 1a). Notably, Zhou and co-workers used 1-iodoalkynes as special internal alkynes to prepare the corresponding chiral 1,4,5-trisubstituted triazoles[24]. In 2021, Topczewski et al. discovered enantioselective nickel-catalyzed click reaction with internal alkynes for dynamic kinetic resolution of allylic azides[40], which gave the expected 1,4,5-trisubstituted triazoles with up to 86% ee (Fig. 1b). Recently, significant progress has been made in the atroposelective click reaction of β-naphthol-derived internal alkynes with azides, independently disclosed by Li[41], Xu[42], and Cui[43,44]. In these protocols, a range of axially chiral 1,4,5-trisubstituted triazoles were prepared by

employing precious rhodium or iridium catalyst (Fig. 1c). Although current methods could provide diverse access to chiral 1,4-disubstituted and 1,4,5-trisubstituted triazoles, there were a few kind of asymmetric click reactions and no reports on efficient strategy for synthesis of chiral 1,5-disubstituted triazoles, which are core units of various bio-active molecules[45–50]. Therefore, exploring a kind of efficient catalytic system to achieve asymmetric click reaction especially for synthesizing chiral 1,5-disubstituted triazoles, is still highly desirable.

Organic azides, as one of the key starting materials for click reactions, have been extensively studied for their asymmetric synthesis[51–59]. Among these, hypervalent iodine-based azidating reagents were often selected as the azido sources[60–66] due to their advantages of mild properties and high selectivity. In previous research, benziodazolone-based azidating reagents were identified as efficient azido transfer reagents that could significantly improve reaction outcomes by adjusting the substituents[64]. In continuation of our interest in hypervalent iodine chemistry[64–69], we report herein an efficient enantioselective copper-catalyzed azidation/click cascade reaction of *N*-propargyl-β-ketoamides and azidobenziodazolone **1d**. This method demonstrates broad substrate scope, easy operation, and mild reaction conditions. Importantly, this work represents catalytic asymmetric example of combining azidation and click reactions to

[1]The Education Ministry Key Laboratory of Resource Chemistry, Joint International Research Laboratory of Resource Chemistry of Ministry of Education, Shanghai Key Laboratory of Rare Earth Functional Materials, and Shanghai Frontiers Science Center of Biomimetic Catalysis, Shanghai Normal University, 200234 Shanghai, China. ✉e-mail: qinghaideng@shnu.edu.cn

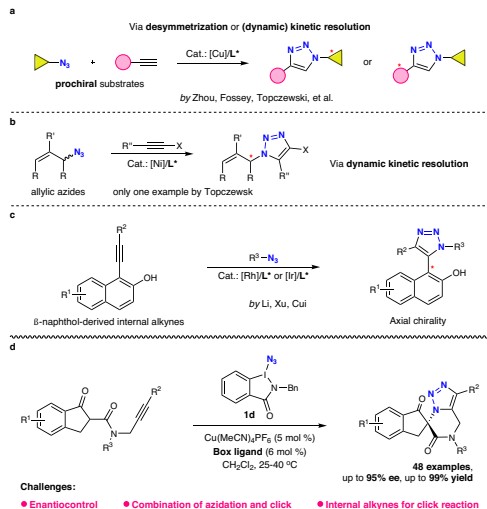

**Fig. 1 | Strategies for catalytic asymmetric synthesis of chiral 1,2,3-triazoles.** **a** Synthesis of chiral 1,4-disubstituted triazoles by enantioselective Cu-catalyzed Click reaction. **b** Synthesis of chiral 1,4,5-trisubstituted triazoles by enantioselective Ni-catalyzed Click reaction. **c** Synthesis of chiral 1,4,5-trisubstituted triazoles by atroposelective Rh/Ir-catalyzed Click reaction. **d** This work: synthesis of chiral 1,5-disubstituted triazoles (unexplored) and 1,4,5-trisubstituted triazoles by enantioselective Cu-catalyzed azidation/Click cascade. Cat. catalyst, L ligand, Cu copper, Ni nickel, Rh rhodium, Ir iridium.

construct chiral triazoles, particularly chiral 1,5-disubstituted triazoles (Fig. 1d).

## Results

We studied the model reaction of *N*-propargyl-β-ketoamide **2a** with azidobenziodoxolone (**1a**) in the presence of the catalyst prepared in situ from 10 mol % of Cu(MeCN)$_4$PF$_6$ and 12 mol % of Box ligand to optimize the reaction conditions (Table 1, see the details of optimization in Supplementary Tables 1–4). When the reaction was carried out in 1,2-dichloroethane at room temperature, the desired chiral triazole **3a** was observed. Screening a series of Box ligands revealed that using **L8** as the optimal ligand afforded **3a** in 62% yield with 92% ee (Table 1, entries 1–10). After evaluating various solvents, it was found that dichloromethane could improve the yield and ee value of **3a** (70% yield, 94% ee; Table 1, entry 11). A range of azidating reagents were subsequently tested. In comparison to **1a** and azidodimethylbenziodoxole (**1b**), azidobenziodazolones (**1c–1g**) were more suitable for the reaction, yielding product **3a** with higher yields even at a reduced catalyst loading of 5 mol % (Table 1, entries 11–17). These results implied that the substituent on the nitrogen atom could significantly influence the reaction efficiency, with *N*-benzyl-azidobenziodazolone (**1d**) identified as the optimal azido source, affording product **3a** in 99% yield with 94% ee (Table 1, entry 14).

With the optimized reaction conditions in hand, we evaluated the scope of the substrates bearing terminal alkyne unit (Fig. 2). When $R^2$ was a benzyl group, all of the substrates were compatible with the

## Table 1 | Optimization of the reaction conditions[a]

| Entry | L | "N$_3$" | Solvent | Time (h) | Yield (%) | ee (%) |
|---|---|---|---|---|---|---|
| 1 | **L1** | **1a** | DCE | 24 | 35 | 14 |
| 2 | **L2** | **1a** | DCE | 24 | 28 | 12 |
| 3 | **L3** | **1a** | DCE | 24 | 50 | 23 |
| 4 | **L4** | **1a** | DCE | 24 | 37 | 75 |
| 5 | **L5** | **1a** | DCE | 24 | 44 | 44 |
| 6 | **L6** | **1a** | DCE | 24 | 40 | 82 |
| 7 | **L7** | **1a** | DCE | 24 | 45 | 80 |
| 8 | **L8** | **1a** | DCE | 24 | 62 | 92 |
| 9 | **L9** | **1a** | DCE | 24 | 47 | 33 |
| 10 | **L10** | **1a** | DCE | 24 | 52 | 31 |
| 11 | **L8** | **1a** | DCM | 24 | 70 | 94 |
| 12[b] | **L8** | **1b** | DCM | 24 | 30 | 27 |
| 13[c] | **L8** | **1c** | DCM | 48 | 80 | 90 |
| 14[c] | **L8** | **1d** | DCM | 48 | 99 | 94 |
| 15[c] | **L8** | **1e** | DCM | 48 | 78 | 94 |
| 16[c] | **L8** | **1f** | DCM | 48 | 95 | 93 |
| 17[c] | **L8** | **1g** | DCM | 48 | 93 | 93 |

DCE: 1,2-dichloroethane, DCM: dichloromethane, L: Ligand; N$_3$: high iodine azide reagent; **1a**: azidobenziodoxolone, **1b**: azidodimethylbenziodoxole; **1c–g**: azidobenziodazolones.
[a]Reaction conditions: **2a** (0.1 mmol), **1a** (0.15 mmol), Cu(MeCN)$_4$PF$_6$ (10 mol%), **L** (12 mol %), solvent (2 mL), 25 °C; isolated yields; ee values were determined by HPLC analysis.
[b]**1b** was used in place of **1a**.
[c]The designated azidating reagent was used in place of **1a**; 5 mol % of Cu(MeCN)$_4$PF$_6$ and 6 mol% of **L8** were used instead.

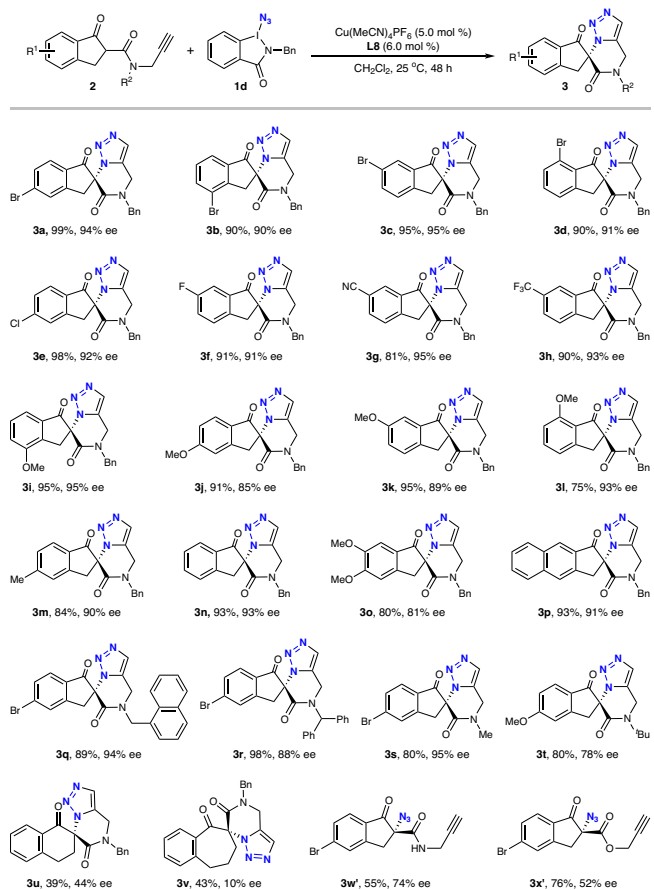

**Fig. 2 | Scope for the synthesis of chiral 1,5-disubstituted triazoles.** Reaction conditions: **2** (0.1 mmol), **1d** (0.15 mmol), Cu(MeCN)₄PF₆ (5 mol%), **L8** (6 mol%), CH₂Cl₂ (2 mL), 25 °C; isolated yields; ee values were determined by HPLC analysis.

reaction conditions to deliver the desired chiral 1,5-disubstituted triazoles **3** in high yields with high to excellent ee values. For example, the substrates bearing diverse substituents at the different sites of the aryl ring, including electron-withdrawing groups (5-Br, 4-Br, 6-Br, 7-Br, 5-Cl, 6-F, 6-CN, 6-CF₃), electron-donating groups (4-OMe, 5-OMe, 6-OMe, 7-OMe, 5-Me) and electron-neutral group (H) were well tolerated under the conditions, affording products **3a–3n** in 75–99% yields and 85–95% ees. Moreover, substrate possessing 4,5-di-OMe on the aryl ring and naphthalene-derived substrate were also successfully employed in the process, generating **3o** (81% yield, 80% ee) and **3p** (93% yield, 91% ee), respectively. Subsequently, the influence of the *N*-substituent (R²) was investigated. Groups such as 1-naphthylmethyl, 1,1-diphenylmethyl, and even methyl were tolerated in this protocol, leading to the formation of the corresponding products **3q–3s** in 80–98% yields with 88–95% ees, while *tert*-butyl amide only gave the product **3t** with 78% ee.

When six-membered ring β-ketoamide **2u** and seven-membered ring β-ketoamide **2v** were used as substrates, the corresponding product **3u** and **3v** were obtained with poor ee values, respectively. Unprotected *N*-propargyl-β-ketoamide **2w** and *O*-propargyl-β-ketoester **2x** were not compatible with the reaction, only generating the corresponding azide **3w'** and **3x'**, respectively. These results showed the limitations of substrate scope.

As is well known, copper-catalyzed click reactions are traditionally limited to terminal alkynes[14,17,18], except for a few special examples[24,70,71]. After investigating the scope of substrates bearing terminal alkyne units, we explored the possibilities for the reaction of internal alkynes (Fig. 3). Using **4a** as the substrate, the cycloaddition step indeed did not proceed completely under the standard conditions mentioned above.

Fortunately, the expected 1,4,5-trisubstituted triazoles **5a** could be prepared in 75% yield and 93% ee just by simply raising the temperature to 40 °C and extending the reaction time to 96 h. Substrates with different groups (R¹) in the backbone got the desired products **5b–5e** in 62–70% yields with 83–94% ees.

When the substituents (R²) of the triple bond were aryl rings with diverse functional groups in different positions, the corresponding products **5f–5u** could be formed in 36–75% yields with 90–94% ees. To our delight, the protocol was also applicable to the substrates with heteroaryl frameworks such as 3-pyridyl and 6-quinolyl, affording the expected products **5v** (60% yield, 90% ee) and **5w** (60% yield, 94% ee), respectively. Furthermore, substrates with alkyl units such as methyl, ethyl and cyclohexyl in the triple bond could be converted to the corresponding products **5x–5z** with excellent enantiocontrol. Notably, the absolute configuration of enantiopure **5b**, **5d**, and **5x** was established to be *R* by single-crystal X-ray structure analysis (for details, see Supplementary Notes). The stereochemical assignment of all other products was also made by analogy.

To demonstrate the synthetic utility of this protocol, a gram scale reaction was conducted with **2a** on a 3 mmol scale (1.15 g) under standard conditions, affording the desired product **3a** (1.21 g) in 95% yield without loss of enantioselectivity (Fig. 4a).

A series of control experiments were conducted to gain a preliminary understanding of the reaction mechanism. The reaction was inhibited completely by adding radical scavengers such as 2,2,6,6-tetramethyl-1-piperidinyloxy (TEMPO), 2,6-di-*tert*-butyl-4-methylphenol (BHT), or 1,1-diphenylethene (DPE) (Fig. 4b). These results revealed that the reaction pathway probably involves a radical process. A linear correlation was observed between the ee values of **L8** and the enantioenrichment of the corresponding products (Fig. 4c). The absence of a nonlinear effect in this reaction suggested that a single catalyst is likely involved in the enantiodetermining transition state[72]. To confirm that the reaction is performed sequentially in the steps of azidation and cycloaddition, we terminated the reaction of **4x** with **1d** to get the intermediate azide **6** with 95% ee, which could be converted to the desired product **5x** in 75% yield with 94% ee under standard reaction conditions. However, the cycloaddition step did not proceed effectively under conditions without a catalyst or with only Cu(MeCN)₄PF₆ as the catalyst (Fig. 4d). These results indicated that the Cu/**L8** catalyst probably involved in both steps of the reaction, with the azidation being the enantio-determining step. To verify the role of the triple bond in enantiocontrol, several control compounds were chosen to perform the asymmetric azidation reaction (Fig. 4e). Ketoester **7a** and ketoamide **7b**, both with no propargyl group, got the corresponding products **8a–8b** with moderate ee values. Compounds **7c** and **7d**, using propyl instead of propargyl, could get the azidating products **8c–8d** with reduced ee values in comparison with those of the analogous triazoles (**3a** and **3q**), respectively. These data implied that the triple bond in the substrate might behave as a directing group in enantiocontrol of azidation step.

To gain insights into the mechanisms of azido transfer and the origin of the enantioselectivity induced by the Box ligand, we carried out DFT calculations to propose a catalytic model (Fig. 4f). The bidentate ligand **L8**, azido anion, and a carbonyl group of substrate coordinate to the Cu(III) center to form a complex with a distorted tetrahedral coordination geometry as the intermediate. The favored conformation of the intermediate (int-I) exhibits an energy that is 8.34 kcal/mol lower than that of the other side (int-II). This discrepancy is primarily attributed to the steric hindrance caused by the carbonyl group of substrate and the *tert*-butyl group of ligand **L8**. Such hindrance leads to an increase in the angle between the exocyclic amide carbonyl and the five-membered ring from 54° (int-I) to 71° (int-II). The enlargement of this angle results in a deterioration of conjugation, which not only elevates the energy level but also makes the enol

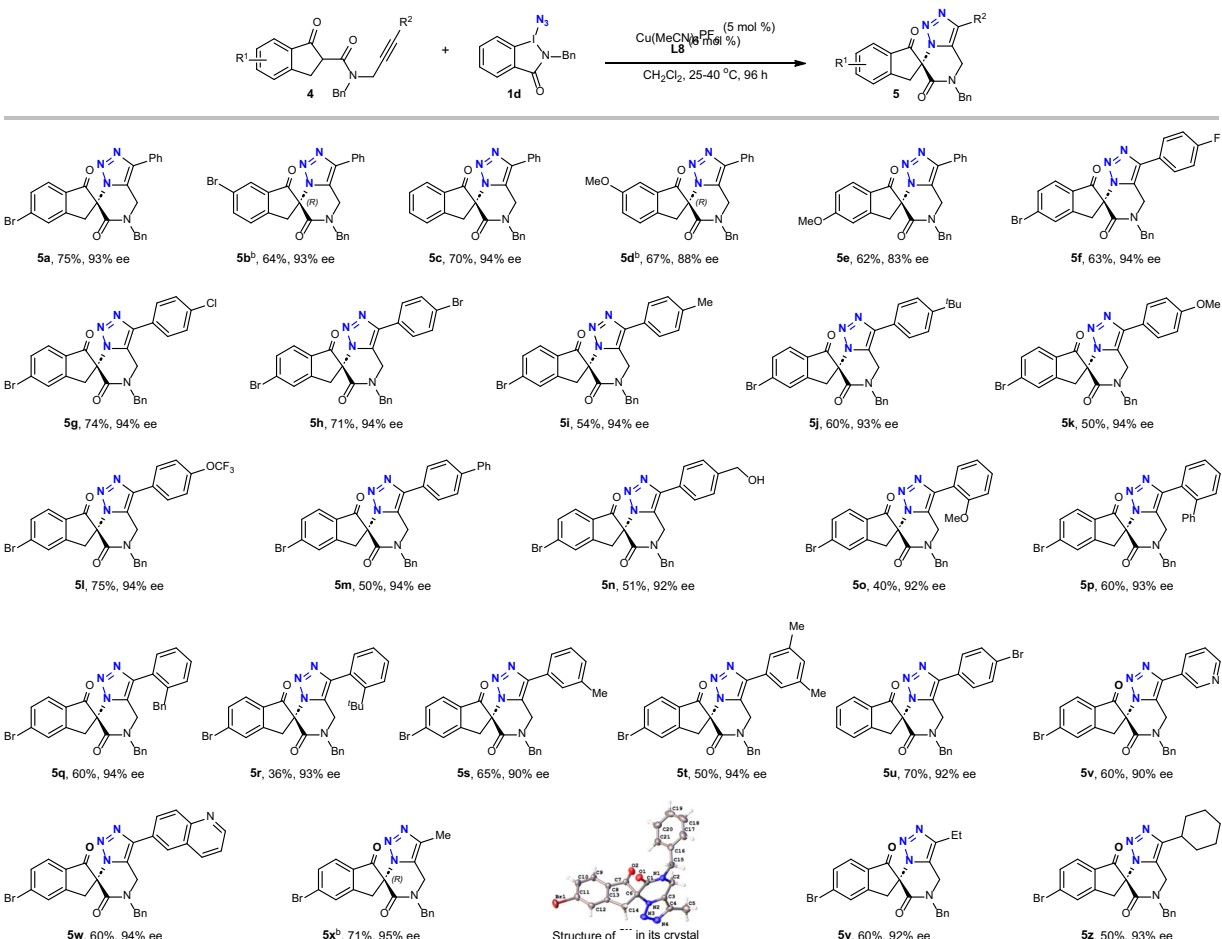

**Fig. 3 | Scope for the synthesis of chiral 1,4,5-trisubstituted triazoles.** Reaction conditions: 4 (0.1 mmol), **1d** (0.15 mmol), Cu(MeCN)₄PF₆ (5 mol %), L8 (6 mol %), CH₂Cl₂ (2 mL); the mixture was stirred at 25 °C for 48 h, then turned to 40 °C for more 48 h; isolated yields; ee values were determined by HPLC analysis. [b]The absolute configuration was established to be *R* by the single-crystal X-ray structure analysis.

double bond more reactive due to the loss of conjugation. Additionally, the enlargement of the angle causes the groups attached to the amide, such as the alkynyl group in this case, to rotate towards the backside of the α-carbon, thereby obstructing the approach of azide.

Even though the exact mechanism remains unclear, a tentative catalytic mechanism was proposed according to the experimental results and previous related reports (Fig. 4f)[52,73,74]. Initially, copper ion coordinates with Box ligand to get the chiral catalyst **A**, which activates azidobenziodazolone **1d** to afford azido-Cu(III) species **B**[73,74]. Intermediate **B** exhibits radical properties and can dissociate into an azido radical. Intermediate **B** then reacts with substrate **4x** to get intermediate **C**, where copper probably coordinates with both a carbonyl group of the substrate and an azido anion. Intermediate **C**, in its favored conformation **int-I**, undergoes stereoselective reductive elimination to afford the chiral intermediate azide **D**. Copper species would coordinate with both the azido group and triple bond in **D** to render intermediate **E**, which probably proceeds the click cycloaddition to form intermediate **F**[75,76]. Finally, **F** releases product **5x** along with the regeneration of catalyst **A**.

## Discussion

In conclusion, we have developed an enantioselective copper-catalyzed azidation/click cascade reaction of *N*-propargyl-β-ketoamides using our previously developed azidobenziodazolone as the efficient azido source. This protocol afforded an efficient and mild route to synthesize chiral triazoles, especially chiral 1,5-disubstituted triazoles, which have not been obtained via reported asymmetric click reactions. Moreover, high functional group tolerance, scalability in synthesis, and easily operated reaction conditions further demonstrated the synthetic utility of this strategy. Initial mechanistic studies revealed that the copper/Box catalyst participates in both azidation and click steps. Further mechanistic investigation and expansion of the azidation/click cascade strategy to access more chiral triazoles are ongoing in our laboratory.

## Methods

### General procedure for the synthesis of racemic 3a–3v

To a mixture of Cu(MeCN)₄PF₆ (0.005 mmol, 5 mol %), substrates **2a–2v** (0.10 mmol), **1d** (0.15 mmol) in dry dichloromethane (2 mL) under nitrogen atmosphere, the reaction system was stirred at 25 °C for 48 h. The disappearance of substrates **2a–2v** was monitored by TLC, indicating complete consumption. Finally, the crude product was purified by silica gel flash chromatography to afford the desired product *rac*-**3a**–*rac*-**3v**.

### General procedure for the synthesis of racemic 5a–5z

To a mixture of Cu(MeCN)₄PF₆ (0.005 mmol, 5 mol %), substrates **4a–4z** (0.10 mmol), **1d** (0.15 mmol) in dry dichloromethane (2 mL) under nitrogen atmosphere, the reaction system was stirred at 40 °C for 48 h. The disappearance of substrates **4a–4z** was monitored by TLC, indicating complete consumption. Finally, the crude product was purified by silica gel flash chromatography to afford the desired products *rac*-**5a**–*rac*-**5z**.

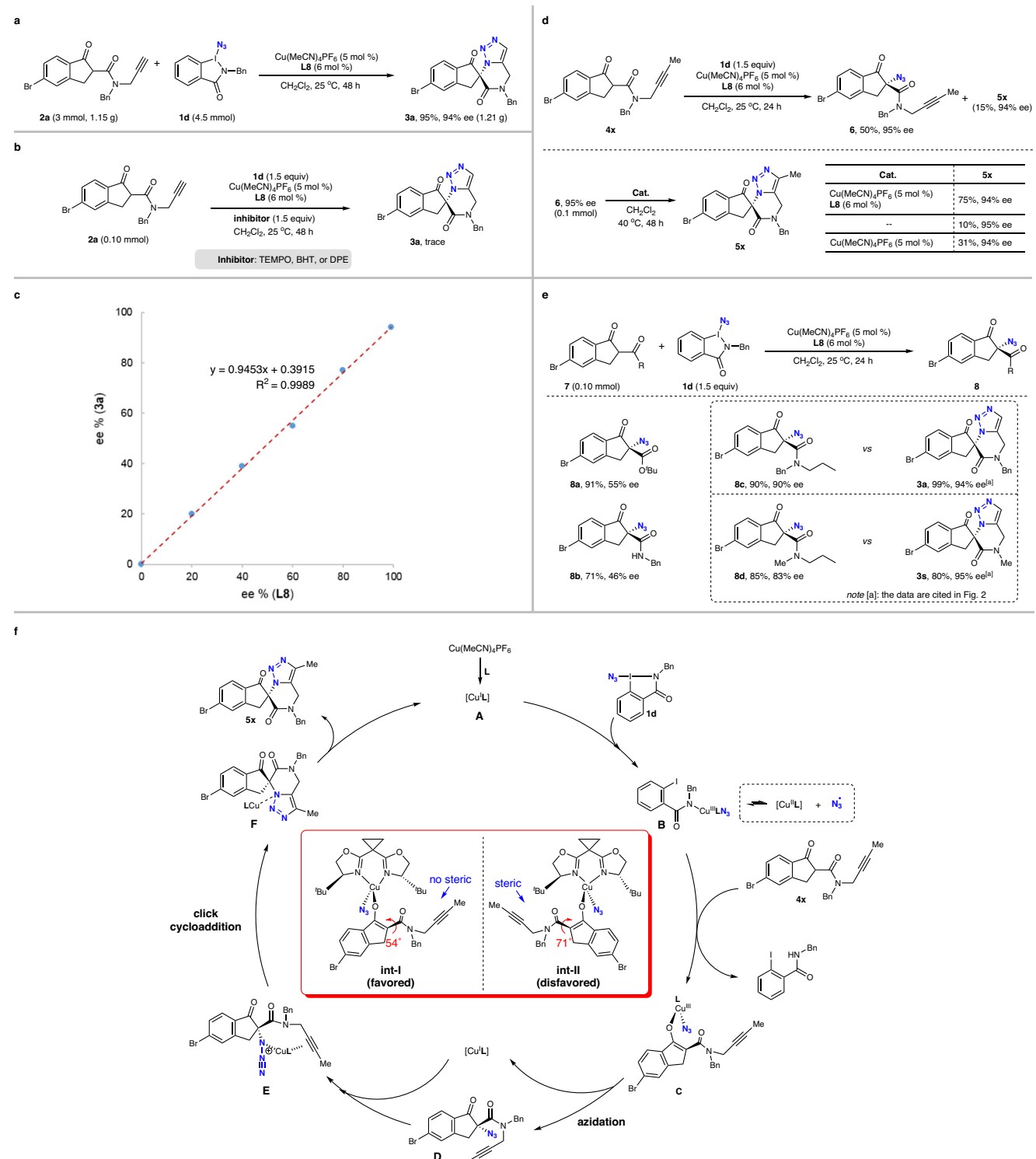

**Fig. 4 | Gram-scale reaction and mechanistic studies. a** Gram-scale reaction. **b** Radical inhibition experiments. **c** Nonlinear effect of catalyst. **d** Investigation of intermediate azide. **e** The role of triple bond in enantiocontrol. **f** Proposed mechanism. TEMPO: 2,2,6,6-tetramethylpiperidoxyl, BHT: butylated hydroxytoluene, DPE: 1,1-diphenylethylene.

## General procedure for the synthesis of chiral 3a–3v

After stirring a mixture of Cu(MeCN)$_4$PF$_6$ (0.005 mmol, 5 mol %) and **L8** (0.006 mmol, 6 mol %) in dry dichloromethane (1 mL) at 25 °C under nitrogen atmosphere for 1.5 h, substrates **2a**–**2v** (0.10 mmol) and **1d** (0.15 mmol) in dry dichloromethane (1 mL) were added. The reaction mixture was then stirred at 25 °C under a nitrogen atmosphere. After 48 h, the substrates **2a**–**2v** disappeared completely

(monitored by TLC). Finally, the crude product was purified by silica gel flash chromatography to afford the desired products **3a**–**3v**.

## General procedure for the synthesis of chiral 5a-5z

After stirring a mixture of Cu(MeCN)$_4$PF$_6$ (0.005 mmol, 5 mol %) and **L8** (0.006 mmol, 6 mol %) in dry dichloromethane (1 mL) at 25 °C under nitrogen atmosphere for 1.5 h, substrates **4a**–**4z** (0.10 mmol)

and **1d** (0.15 mmol) in dry dichloromethane (1 mL) were added. The reaction mixture was stirred at 25 °C under a nitrogen atmosphere. After 48 h, when the substrates **4a**–**4z** disappeared (monitored by TLC), the reaction mixture was stirred at 40 °C for an additional 48 h. Finally, the crude product was purified by silica gel flash chromatography to afford the desired product **5a**–**5z**.

## Data availability

The authors declare that the data supporting the findings of this study, including synthetic procedures, characterization data, further details of computational studies and NMR spectra, are available within the article and the Supplementary Information file, or from the corresponding author upon request. The X-ray crystallographic coordinates for structures reported in this study have been deposited at the Cambridge Crystallographic Data Centre (CCDC) under deposition numbers CCDC2327853 (for **5b**), CCDC2327852 (for **5d**), and CCDC2327851 (for **5x**). These data can be obtained free of charge from The Cambridge Crystallographic Data Centre via https://www.ccdc.cam.ac.uk/structures/. Source data are provided with this paper.

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

## Acknowledgements

This paper is in memory of Professor Lixin Dai. We are grateful for the financial support from the Shuguang program (20SG44) from Shanghai Education Development Foundation and Shanghai Municipal Education Commission, the National Natural Science Foundation of China (22371187), the Natural Science Foundation of Shanghai (22ZR1445200), the Chinese Education Ministry Key Laboratory and International Joint Laboratory on Resource Chemistry, the "111" Innovation and Talent Recruitment Base on Photochemical and Energy Materials (D18020), and the Shanghai Engineering Research Center of Green Energy Chemical Engineering (18DZ2254200).

## Author contributions

Q.-H.D. conceived and directed the project. L.-F.J. and S.-H.W. conducted most of the experiments, including the synthesis of the hypervalent iodine azidating reagents and substrates. Y.-X.J., H.-X.M., J.-J.H., Y.-B.B., D.-Y.K., Y.-F.C., and X.C. synthesized some substrates. L.-F.J. performed the computational studies. L.-F.J. drafted the Supporting Information. Q.-H.D. prepared the manuscript and revised the Supporting Information.

## Competing interests

The authors declare no competing interests.
