## [Peer Review File · Nature Communications]

Enantioselective copper-catalyzed azidation/click cascade:
easy access to chiral 1,2,3-triazolesREVIEWER COMMENTS

Reviewer #1 (Remarks to the Author):

Deng and coworkers report a nice work of Cu-catalyzed asymmetric azidation/cyclization cascade to achieve chiral triazoles. The Cu catalyst works in the both two steps, first to incorporate azide functionality with enantioselectivity control, and then promote the cyclization. It seems that the ligand plays a vital role in the process. Thus a lot of chiral triazoles were prepared with broad diversity, and I think this work would receive much attention in medicinal chemistry. Overall, a nice work well conducted and well organized. I suggest a minor revision before publication.

(1) There exists some typos, for example, page 12, line 222. "Tafter stirring a mixture" should be "To a stirring mixture"; page 6, line 101, "electron-netural" should be "electron-neutral".

(2) In page 7, the chemical name in figure 3's legend should be bold. for example, 4, 1d, L8. Please check all in the text.

(3) In page 7, figure 3, all the N atom in 5i should be labeled to blue color to keep consistent.

(4) In page 10, figure 4. A well conducted mechanism investigation. However, in (f), the mechanism proposed section. From D to 5i, I think Cu catalyst would coordinate both with azide and alkyne to render a cyclization transition state. Just as in CuAAC, Cu coordinate with azide and alkyne.

(5) In page 3, figure 1d. This atroposelective click reaction has an original non-chiral version, *Adv. Synth. Catal.* 2019, 361, 989–994. This could be cited.

Reviewer #2 (Remarks to the Author):

The author Qing-Hai Deng et al. reported an enantioselective azide-alkyne click cyclisation to afford chiral 1,2,3-triazoles with high yield and ee under mild conditions. They also were able to report the synthesis of chiral 1,5-disubstituted triazoles.

The paper has its own merits though similar works towards the synthesis of chiral triazoles have been reported. However, the following points need clarification/investigation before its publication.

1. Full English language check is needed.

2. The role of chiral ligand L8 is not clear. Probably Cu (I)-gets attached which should have more affinity to coordinate with 1,3-diketone unit followed by azide association in the position shown. Thus, details understanding should be presented.

3. Obviously there must be activation of the alkyne by the Cu (I). However, this may not be sole cause of chiral azidation.

4. How the radical proposition helps in enantioselectivity?

5. Once chiral azide is being formed and isolated whether the authors have tried the click reaction with simple Cu(I) or not is not mentioned. In that case whether there is any role of the chiral catalyst L8?

6. The azide is too crowded. How facile the reaction is?

7. How wise is to consider the involvement of Cu(III)?

8. Whether the same azidation can go with other azide transferring reagent?

Reviewer #3 (Remarks to the Author):

In this work, Deng and co-workers report an example of copper catalyzed enantioselective

azidation/click cascade reaction to access a range of chiral 1,2,3-triazoles containing a spiro carbon stereocenter. Despite previous reports of asymmetric click reactions, the synthesis of structurally complex triazoles by sequential asymmetric azidation and click reactions remains high novelty. In particular, chiral 1,5-disubstituted triazoles that have not been accessed by previous methods were also prepared with good functional group tolerance. The key aspects of this work lie in the combination of two-step reactions through substrate design and the use of suitable azidating reagents reported before by the authors. A series of control experiments were conducted to help understand the role of copper complex in the catalytic cycle. The reviewer would support publication of this work in Nature Communications after minor revision.

Additional points:

- 1) The method is tolerated with N-propargyl- β -ketoamides. The authors stated that the N-propargyl group is crucial to the enantioselectivity of the reaction. What's the result if the N-propargyl group is replaced with O-propargyl? How about unprotected N-propargyl- β -ketoamides as the substrates? In addition, a range of cyclic five-membered ring β -ketoamides were examined. How about six-membered ring β -ketoamides or acyclic substrates? The information should be included at least in SI to help understand the scope and limitations of the method.
- 2) Did the authors observe the axial chirality in products 5o-5q? Suggest the authors try the substrates with bulker group (e.g.: tBu) in the ortho- position of aryl ring (R2) to observe if any axially chiral products are formed.
- 3) In SI, The authors should zoom in on the regions with too many peaks in the ^1H NMR and ^{13}C NMR spectra of 2q, 4p, 4q, etc.

Our responses to the reviewers' comments

REVIEWER COMMENTS

Reviewer #1 (Remarks to the Author):

Deng and coworkers report a nice work of Cu-catalyzed asymmetric azidation/cyclization cascade to achieve chiral triazoles. The Cu catalyst works in the both two steps, first to incorporate azide functionality with enantioselectivity control, and then promote the cyclization. It seems that the ligand plays a vital role in the process. Thus a lot of chiral triazoles were prepared with broad diversity, and I think this work would receive much attention in medicinal chemistry. Overall, a nice work well conducted and well organized. I suggest a minor revision before publication.

(1) There exists some typos, for example, page 12, line 222. "After stirring a mixture" should be "To a stirring mixture"; page 6, line 101, "electron-neutral" should be "electron-neutral".

Our response: Thanks for your kind reminder. We corrected these mistakes in the revised manuscript.

(2) In page 7, the chemical name in figure 3's legend should be bold. For example, 4, 1d, L8. Please check all in the text.

Our response: Thanks for your kind reminder. We corrected these mistakes in the revised manuscript.

(3) In page 7, figure 3, all the N atom in 5i should be labeled to blue color to keep consistent.

Our response: Thanks for your kind reminder. We corrected these mistakes in the revised manuscript.

(4) In page 10, figure 4. A well conducted mechanism investigation. However, in (f), the mechanism proposed section. From D to 5i, I think Cu catalyst would coordinate both with azide and alkyne to render a cyclization transition state. Just as in CuAAC, Cu coordinate with azide and alkyne.

Our response: Thanks for your kind suggestion. We adjusted the proposed mechanism in the revised manuscript:

Even though the exact mechanism remains unclear, a tentative catalytic mechanism was proposed according to the experimental results and previous related reports (Fig. 4f)^{52, 73, 74}. Initially, copper ion coordinates with Box ligand to get the chiral catalyst **A**, which activates azidobenziodazolone **1d** to afford azido-Cu(III) species **B**^{73, 74}. Intermediate **B** exhibits radical property and can dissociate into an azido radical. Intermediate **B** then reacts with substrate **4x** to get intermediate **C**, where copper probably coordinates with both a carbonyl group of substrate and an azido anion. Intermediate **C**, in its favored conformation **int-I**, undergoes stereoselective reductive elimination to afford the chiral intermediate azide **D**. Copper species would coordinate with both the azido group and triple bond in **D** to render intermediate **E**, which probably proceeds the click cycloaddition to form intermediate **F**^{75, 76}. Finally, **F** releases product **5x** along with the regeneration of catalyst **A**.

(5) In page 3, figure 1d. This atroposelective click reaction has an original non-chiral version, *Adv. Synth. Catal.* 2019, 361, 989–994. This could be cited.

Our response: Thanks for your kind reminder. We cited this reference as ref. 43.

Reviewer #2 (Remarks to the Author):

The author Qing-Hai Deng et al. reported an enantioselective azide-alkyne click cyclisation to afforded chiral 1,2,3-triazoles with high yield and ee under mild

conditions. They also were able to report the synthesis of chiral 1,5-disubstituted triazoles.

The paper has its own merits though similar works towards the synthesis of chiral triazoles have been reported. However, the following points need clarification/investigation before its publication.

1. Full English language check is needed.

Our response: Thanks for your kind reminder. We corrected some grammatical errors and typos in the revised manuscript.

2. The role of chiral ligand L8 is not clear. Probably Cu (I)-gets attached which should have more affinity to coordinate with 1,3-diketonic unit followed by azide association in the position shown. Thus, details understanding should be presented.

Our response: Thanks for your kind suggestion. To gain insights into the role of chiral ligand **L8**. We carried out DFT calculations to propose a catalytic model, and added the descriptions in the revised manuscript:

To gain insights into the mechanisms of azido transfer and the origin of the enantioselectivity induced by the Box ligand, we carried out DFT calculations to propose a catalytic model (Fig. 4f). The bidentate ligand **L8**, azido anion, and a carbonyl group of substrate coordinate to the Cu(III) center to form a complex with a distorted tetrahedral coordination geometry as the intermediate. The favored conformation of the intermediate (**int-I**) exhibits an energy that is 8.34 kcal/mol lower than that of the other side (**int-II**). This discrepancy is primarily attributed to the steric hindrance caused by the carbonyl group of substrate and the *tert*-butyl group of ligand **L8**. Such hindrance leads to an increase in the angle between the exocyclic amide carbonyl and the five-membered ring from 54° (**int-I**) to 71° (**int-II**). The enlargement of this angle results in a deterioration of conjugation, which not only elevates the energy level but also makes the enol double bond more reactive due to the loss of conjugation. Additionally, the enlargement of the angle causes the groups attached to the amide, such as the alkynyl group in this case, to rotate towards the backside of the α -carbon, thereby obstructing the approach of azide.

3. Obviously there must be activation of the alkyne by the Cu (I). However, this may not be sole cause of chiral azidation.

Our response: Thanks for your kind suggestion. On basis of the results of DFT calculations, it was found that the enantiocontrol for azidation is likely achieved through copper coordinating with a carbonyl group in the substrate, while the alkyne group tends to provide steric hindrance rather than coordinating with copper to control enantioselectivity. Therefore, we modified the mechanism shown in Fig. 4f. Please also see the response mentioned as above.

4. How the radical proposition helps in enantioselectivity?

Our response: We inferred that the reaction likely proceeds via a radical pathway on basis of the results of radical inhibition experiments (Fig. 4b in the manuscript). This pathway has been extensively studied and confirmed as a plausible route for achieving chirality control in various copper-catalyzed asymmetric reactions.

5. Once chiral azide is being formed and isolated whether the authors have tried the click reaction with simple Cu(I) or not is not mentioned. In that case whether there is any role of the chiral catalyst **L8**?

Our response: Thanks for your kind suggestion. We have investigated the reaction conditions for the click reaction of chiral intermediate azide **6** (Please see Fig. 4d in the manuscript) and have inferred that the reaction likely proceeds through a radical pathway on basis of the results of radical inhibition experiments (Fig. 4b in the manuscript). “..... we terminated the reaction of **4x** with **1d** to get the intermediate azide **6** with 95% ee, which could be converted to the desired product **5x** in 75% yield with 94% ee under standard reaction conditions. However, the cycloaddition step did not proceed effectively under conditions without catalyst or with only Cu(MeCN)₄PF₆

as the catalyst (Fig. 4d).” These results indicated that the Cu/L8 catalyst probably also involves in the click reaction. We speculate that L8 may promote the click reaction by altering the electronic properties of the copper center.

6. The azide is too crowded. How facile the reaction is?

Our response: For terminal alkyne substrates, the click reactions were indeed rapid, making it difficult to isolate the corresponding pure azide intermediates. For non-terminal alkyne substrates, the click reactions were relatively more challenging. We were able to isolate the corresponding azide intermediate **6**, and the subsequent click reaction proceeded more efficiently only catalyzed by Cu/L8 complex (Please see Fig. 4).

7. How wise is to consider the involvement of Cu(III)?

Our response: Cu(III) species are commonly encountered as reaction intermediates in reactions involving hypervalent iodine reagents. They are typically generated through the oxidative addition of Cu(I) with hypervalent iodine reagents. Based on experimental evidence and previous literatures (refs. 52 and 73), the formation of Cu(III) species is considered plausible.

8. Whether the same azidation can go with other azide transferring reagent?

Our response: Thanks for your kind suggestion. We screened other azide transferring reagents, for example, TMSN₃/oxidants, which were shown in Table S4. However, compound **1d** is still the optimal azido source.

Reviewer #3 (Remarks to the Author):

In this work, Deng and co-workers report an example of copper catalyzed enantioselective azidation/click cascade reaction to access a range of chiral 1,2,3-triazoles containing a spiro carbon stereocenter. Despite previous reports of asymmetric click reactions, the synthesis of structurally complex triazoles by sequential asymmetric azidation and click reactions remains high novelty. In particular, chiral 1,5-disubstituted triazoles that have not been accessed by previous methods were also prepared with good functional group tolerance. The key aspects of this work lie in the

combination of two-step reactions through substrate design and the use of suitable azidating reagents reported before by the authors. A series of control experiments were conducted to help understand the role of copper complex in the catalytic cycle. The reviewer would support publication of this work in Nature Communications after minor revision.

Additional points:

1) The method is tolerated with *N*-propargyl- β -ketoamides. The authors stated that the *N*-propargyl group is crucial to the enantioselectivity of the reaction. What's the result if the *N*-propargyl group is replaced with *O*-propargyl? How about unprotected *N*-propargyl- β -ketoamides as the substrates? In addition, a range of cyclic five-membered ring β -ketoamides were examined. How about six-membered ring β -ketoamides or acyclic substrates? The information should be included at least in SI to help understand the scope and limitations of the method.

Our response: Thanks for your kind suggestion. When six-membered ring β -ketoamide **2u** and seven-membered ring β -ketoamide **2v** were used as the substrates, the corresponding product **3u** and **3v** were obtained with poor ee values, respectively. Unprotected *N*-propargyl- β -ketoamide **2w** and *O*-propargyl- β -ketoester **2x** were not compatible with the reaction, only generating the corresponding azide **3w'** and **3x'**, respectively. These results showed the limitations of substrate scope. (Please also see the descriptions in the revised manuscript and Fig. 2)

In addition, we were unable to get pure acyclic substrate **2y**, because it was easily converted to the cyclic isomer **S11**. (Please also see page S8 in SI)

2) Did the authors observe the axial chirality in products 5o-5q? Suggest the authors try the substrates with bulkier group (e.g.: ^tBu) in the ortho- position of aryl ring (R²) to observe if any axially chiral products are formed.

Our response: Thanks for your kind suggestion. We also synthesized the substrate **4r** bearing *o*-^tBu-Ph group, which proceeded the reaction to get the desired product **5r** in 36% yield with 93% ee. Notably, we did not observe the axial chirality in products **5o-5r**. (Please also see Fig. 3 in the revised manuscript)

3) In SI, The authors should zoom in on the regions with too many peaks in the ¹H NMR and ¹³C NMR spectra of 2q, 4p, 4q, etc.

Our response: Thanks for your kind reminder. We modified and updated these spectra.

REVIEWERS' COMMENTS

Reviewer #1 (Remarks to the Author):

The author has addressed all the issues raised previously and this paper could be accepted in this version.

Reviewer #2 (Remarks to the Author):

It may now be accepted for a publication.